# The Impact of Nomophobia, Stress, and Loneliness on Smartphone Addiction among Young Adults during and after the COVID-19 Pandemic: An Israeli Case Analysis

Moti Zwilling 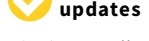

Department of Economics and Business Administration, Ariel University, Ariel 40700, Israel; motiz@ariel.ac.il

**Abstract:** Technological development in recent times has dramatically changed the way people live, interact with, and consume information. Since the emergence of the first iPhone in January 2007 until today, mobile phones are used daily for a range of purposes. Using mobile phones for various purposes intensified during the COVID-19 pandemic due to isolation or government lockdown regulations. However, along with the advantages of smartphone usage there are many disadvantages such as smartphone addiction and continuous exposure to digital screens, behaviors known as PSU—problematic smartphone use. This study explores the impact of several variables on PSU: loneliness, the need for social interaction, sleep hours, fear of losing phone access (nomophobia), and stress among young adults in Israel. The variables are examined with respect to two time periods: 1. During the COVID-19 pandemic lockdowns (defined as "T1") and 2. Following the end of the first wave of COVID-19 lockdowns (defined as "T2"). The results indicated that nomophobia, social affiliation, and sleep hours affect PSU. In addition, the indirect effect of the number of sleep hours on stress and PSU was found to be significant in T2 and in T1. The findings and their implications are discussed.

**Keywords:** stress; nomophobia; social engagement; loneliness; mobile addiction

## 1. Introduction

Technology, in general and mobile phones, in particular, have in recent years become an inseparable part of our lives. The purposes of using mobile phones range from people's need to search for all kinds of information relevant to daily activities [1] to their desire to take part in discussions on social media [2], and to find information related to their health. When the COVID-19 pandemic erupted in March 2020, in many countries around the world and in Israel, people were suddenly forced to stay at home due to lockdown policies. Many services, including health services, were provided mainly, but not exclusively, via remote mobile applications, also referred to as "smartphone apps" to people with and without familiarity in handling technologies.

The smartphone has become an integral part of many people's lives and is considered a device without which it is not possible to manage in daily life [3,4]. As a device that can immediately provide access to information, news, digital health services, and remote classes for students of all ages, there are diverse advantages of smartphone use. Another advantage is, for example, an increase in the frequency of social media use to assist in maintaining existing relationships as well as in forming new ones [5]. However, the smartphone, which is also used as a channel for obtaining credible information [6], is perceived to have downsides [7,8]. Mobile overuse may lead to negative health outcomes such as addiction, which can be a chronic condition that not only refers to dependence on drug substances but is characterized by the inability to stop partaking in activities such as online gaming, interactions, etc. People with addictions may develop behavior and symptoms that become compulsive and often continue despite their harmful consequences [9] and health symptoms that negatively affect mental and physical health [10].

The use of the term "addiction" when it comes to mobile phones is considered controversial among researchers, but the fact that the term can be associated with excessive viewing on television, or excessive internet browsing, laid the basis for using the term, as applied to mobile phone use as well. Various studies have shown that increased or excessive use of a mobile phone can lead to severe sleep disturbances, psychological distress, economic problems, and loneliness as well [11].

The increased, excessive, and problematic use of smartphones among many people around the world in recent years was accelerated not solely, but also, due to the substantial impact of social networks [12]. Social networks, in addition to cultural and social needs, allow individuals to engage in social activities using computers or smartphones, to track their activities in virtual communities and to interact with other people. The use of social media has increased worldwide, and social media, including Facebook, Twitter, and Instagram, has led users to experience symptoms of addiction.

The ongoing COVID-19 pandemic has led to a significant impact on the personal lifestyles and health of many people. Due to government policies to limit the spread of the virus people, willingly or not, were forced to shift their offline daily activities, such as shopping, to online [13,14]. Other activities included remote work, remote learning via Zoom™ or Microsoft Teams™, as well as remotely delivered health services. The side effect of staying at home was the elimination of face-to-face social interaction; hence, many people who were not spending much time or not exposed at all to social networks and mobile use suddenly found themselves spending more time on social media sites such as Facebook and Twitter for various purposes such as, seeking information related to the pandemic as well as for entertainment and interpersonal communication. The overuse of smartphones and social media is reported especially, but not solely, among youth and young adults as one of the triggers for an increase in the level of stress and addiction along with fewer sleep hours and inactivity [15].

Apart from stress, loneliness was also found to be a risk factor for many people especially those who are constantly connected to social media, who experience discomfort if they are not continuously connected. This causes them to be uninvolved and neglect activities not related to social media. The overuse of social media or mobile phones is also defined as being a major factor in the number of hours of sleep and their quality. For example, online activity (e.g., chat) just before bedtime can cause emotional, mental, and/or physiological arousal, which may, in turn, lead to greater difficulty in falling asleep and result in poor sleep quality [16,17]. In addition, it was found that one of the modern world's disorders, already manifest among young people, is called nomophobia, i.e., the fear of not being able to access and enjoy information through mobile phones [18]. The impact of these factors on the level of smartphone addiction due to the COVID-19 pandemic was not fully addressed in the literature. For this reason, the purpose of this study was to explore the level of smartphone addiction among Israeli subjects, and specifically young adults, during two time periods: during the COVID-19 lockdowns (time period—T1) and afterwards (time period—T2) and examine addiction as a function of the above-described factors. The study's implications are discussed.

## 2. Conceptual Framework—Literature Review

The outbreak of the COVID-19 pandemic occurred in China around December 2019. The rapid spread of the virus around the world forced many governments to adopt lockdown and social distancing polices and to strictly enforce stay at home orders and regulations. As a result, in order to keep in touch with friends and relatives and carry out other aspects of daily life, people were compelled to spend more time on the internet. These COVID-19-induced changes were found to have a negative psychological and behavioral impact especially on young people [19]. The impact was expressed in two ways: 1. Psychological, such as feeling lonely, being stressed, or having a phobia of not being able to communicate with others as well as being detached from mobile phone connectivity. 2. Behavioral, where people were eager to, for example, develop social affiliations with oth-

ers, to belong to a social community, or a tendency to sleep fewer hours than normally. Each of the psychological and behavioral elements, on its own, has been found to be connected to excessive technology usage especially of mobile phones, which may lead to problematic smartphone use (termed "PSU"), defined as an addictive behavior. The psychological and behavioral aspects are listed below:

### 2.1. Psychological Influences

1.  Loneliness: Loneliness is defined as one of the most prevalent global problems for adults. It is considered a "pervasive and adverse psychological state with the feeling of emotional isolation state of being alone and separation from others" ([20], p. 1). Such feeling may lead to increased mortality and other health risks [21]. For example, loneliness is linked to clinical diseases such as stroke and cardiovascular illness, and it is also a predictor of psychological symptoms such as depression, stress, and anxiety [22]. People who feel lonely have been found to be more likely to use their smartphones in an extreme manner for social purposes, tending to use social media platforms in a way akin to an addictive behavior [23,24]. However, several studies (e. g., Skues et al. [25]) have found that loneliness is not a significant predictor of PSU. Therefore, such association requires further exploration.

2.  Stress: The first attempt to define stress occurred in 1859 by Claude Bernard [26]. Many years later in 1926, Walter Bradford Cannon [27] defined stress as a condition in which an organism reacts to threat in a way that impairs its homeostatic equilibrium [28]. According to DSM-5, stress is defined by two disorders: acute stress disorder and posttraumatic stress disorder. Acute stress, where a subject experiences trauma during or after an event, is associated with a sense of numbing and a reduction in awareness of surroundings, which often lead to impairment in social interactions. Already in 1984, Lazarus and Folkman [29] distinguished between two types of coping strategies that subjects typically used to manage stress: 1. Focus on the problem (e.g., seeking health-related information relevant to their stress) and 2. Emotionally focused coping (e.g., venting emotions to manage their mood and joining social support communities). Recently, another study, conducted by Zhao and Zhou [15], showed the same phenomenon: people who experienced stress during the COVID-19 pandemic tended to be more active on social media communities and were at high risk for technology addiction. Similarly, Gao et al. [30] showed that individuals who lose control of their emotional cognizance may express a weakened emotional adjustment that might result in them not being able to cope with difficult and stressful situations. This may not only exacerbate their negative emotions but also increase the likelihood of them developing a severe addiction to their mobile phones.

3.  Nomophobia: By 2013, nomophobia was already considered a modern disorder [31]. Yildirim and Correia [18] described it as a phenomenon whose dimensions include being anxious about losing communication with others, not being able to access to information through their phones, and not having the convenience of access to smartphone applications. Later, Bhattacharya et al. [32] defined nomophobia as a psychological condition where "people are afraid of being detached from mobile phone connectivity" ([32], p. 1297). This definition has been proposed for inclusion as a psychological disorder in the fifth edition of the American Diagnostic and Statistical Manual of Mental Disorders (DSM-5) [33].

### 2.2. Behavioral Influences

1.  Desire to belong to a social community: social media plays a useful role in interpersonal communication. The need to make social media accessible to more people and enhance their user experience influenced the design of such applications, especially on mobile phones, such that due to instant rewards (likes and re-tweets), these social platforms become more addictive [34].

2. Lack of a sufficient number of sleep hours: The need for a sufficient number of sleep hours has already been shown to be of great importance to optimal health and wellbeing [35]. Not getting enough sleep hours may lead to changes in behavior [36] and is associated with attention problems, poor academic performance, daytime fatigue, depression, and obesity [37]. The literature includes many studies that explore the different aspects related to the quality of sleep hours and its connection to technology usage. Hasanzade [38], for example, found that pathological, excessive use of the Internet is correlated with an insufficient number of sleep hours.

   The combination of the described psychological and behavioral manifestations has been found to be related to PSU and to take the form of the following addictive behaviors:

3. PSU and smartphone addiction: Smart mobile phones, also known as smartphones, have quickly become a staple of daily life for many people, especially among younger people, who were found to be more dependent on them. This dependency has led to a new phenomenon: smartphone addiction. This addiction, in general, refers to a situation of uncontrolled and excessive use of a smartphone. For example, Tateno et al. [39] found that in Japan, young people have their smartphones within reach almost all day, and the frequency of using them for Internet browsing and connecting to others is continuously increasing. This is also the case globally and is expected to continue. While other studies [40–42] have found differences between genders when it comes to problematic Internet and smartphone use, both men and women who suffer from low self-esteem, loneliness, depression, interpersonal anxiety, and tend not to belong to social groups exhibit a high level of dependency on their smartphones [43]. Another study conducted by Arpaci [44], which aimed to examine the relationship between social anxiety, smartphone use, a tendency toward trust, and problematic smartphone use showed that smartphone users who tend to rely on others exhibit a high level of problematic smartphone use.

Addiction to technology, in general, and to smartphones, in particular, has recently led to studies that test additional factors that may be correlated with the problematic use of smartphones among adults, young people, and adolescents. For example, Csibi et al. [45] explored the differences in risk of smartphone-related addictive behavior among different ages and found that adults in different age groups show evidence of addictive behavior including the salience of phone use and withdrawal symptoms when unable to use the smartphone. Another interesting aspect of smartphone addiction is related to a recent phenomenon termed "phubbing," a relatively new social phenomenon that refers to cases when a smartphone user ignores other people with whom he is involved in face-to-face interaction, preferring their smartphone [46]. While it may seem that such behavior is normal and even harmless, recent studies point to the fact that such behavior may lead to a disconnection from other people and, finally, to the loss of interpersonal communication. This phenomenon may be accompanied by an increase in smartphone addiction [47].

The effect some of the individual variables on PSU has been explored in the past. The present study wished to examine how a group of variables, as a whole, affect PSU. Accordingly, the study proposes the following hypotheses:

**Hypothesis 1 (H1).** *Loneliness is positively associated with the desire to develop greater social affiliation.*

**Hypothesis 2 (H2).** *The desire to develop increased social affiliations is positively associated with PSU.*

**Hypothesis 3 (H3).** *Social affiliation (throughout this article, the term "social affiliation" will be used as shorthand for "the desire to develop greater social affiliations.") will serve as a mediator between loneliness and PSU.*

**Hypothesis 4 (H4).** *Nomophobia is positively associated with PSU; hence, mobile users with higher nomophobia levels will exhibit higher levels of PSU.*

**Hypothesis 5 (H5).** *Mobile users with fewer sleep hours will exhibit a higher level of PSU.*

**Hypothesis 6 (H6).** *Stress level will negatively correlate with the number of sleep hours.*

**Hypothesis 7 (H7).** *Sleep hours will serve as a mediator between stress and PSU, i.e., subjects with high levels of stress will tend to have a higher reduction in sleeping hours and therefore exhibit high levels of PSU as compared to individuals who have lower levels of stress and a higher number of sleeping hours.*

Summarizing the above, the study proposed the following hypothesized model (Figure 1):

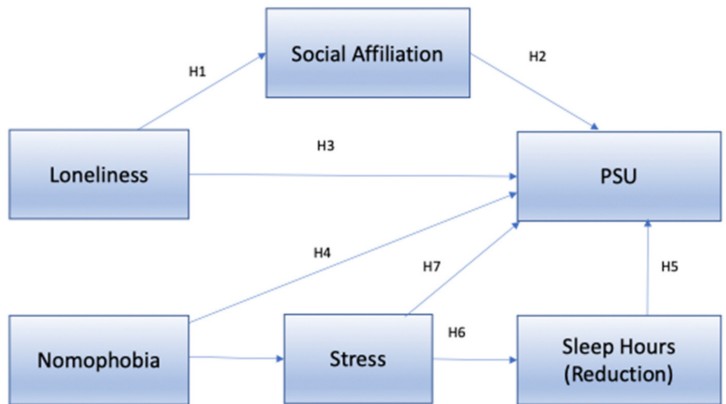

**Figure 1.** Hypothesized model (Study 2).

### 3. Materials and Methods

#### 3.1. Subjects and Sample

An online anonymous questionnaire created on Google Forms™ was initially distributed to a convenient sample—of undergraduate and graduate students in the Department of Economics and Business Administration at Ariel University in Israel (*n* = 207). This phase, hereinafter referred to as Study 1, was defined as an exploratory pretest research. In the next phase (confirmatory trial, hereinafter Study 2), the questionnaire was distributed to 246 respondents aged 18–35 recruited from the general public. This trial was conducted by i-panel™, the largest and leading panel and survey management company in Israel. Regarding the Study 1 process, ten students did not complete or refused to complete the questionnaire (attrition rate = 4.51%); thus, the total number of participants was 197; 54% male (*n* = 106) and 46% female (*n* = 91). Of the respondents, 76% (*n* = 149) already had an academic diploma and 24% (*n* = 48) did not (M_education = 2.31; *s.d.* = 0.78). The average age of respondents who combine work and study was 38 (*s.d.* = 1.18) All respondents possessed a smartphone. As related to the study conducted among the general public: All the participants completed the questionnaire; thus, the total number of participants was 246; 32% male (*n* = 81) and 67% female (*n* = 165). Of the respondents, 57.7% (*n* = 142) already had an academic diploma and 42.3% (*n* = 104) did not (M_education = 3.35; *s.d.* = 0.83). The average age of respondents was 27.41 (*s.d.* = 5.96) All respondents possessed a smartphone.

#### 3.2. Instruments and Measures

To test the conceptual framework of the research, an online questionnaire consisting of closed questions for the six explored variables was created on Google Forms™. The questionnaire had two parts consisting of identical questions. Each part asked the respondents about a significant time period: a. The COVID-19 lockdowns (T1—March to June 2020) and

b. After the COVID-19 lockdowns ended (T2—July to August 2020). The questionnaires were administered in June 2021.

The questionnaire contained six variables with five questions (items) for each variable examined, based on existing validated questions adopted from the literature for each explored variable. Although all the variables and associated questions in the questionnaire were employed according to the literature scales, a convergent validity and internal consistency reliability test of the questionnaire was carried out on Study 1 and Study 2 data. All the variables' questions were translated and adapted to the Hebrew language, and the original scales were coded respectively according to their format in the literature. The following variables were constructed in the questionnaire: 1. loneliness, 2. social affiliation, 3. nomophobia, 4. stress, 5. number of sleep hours, and 6. problematic smartphone use. Specifically, for each evaluated variable the following scales were adapted: 1. Loneliness was measured on a 4-point Likert scale (1—Never . . . 4—Always) adapted from Russell [48]; ($\alpha = 0.84$). 2. Social affiliation was measured on a 5-point Likert scale (1—Definitely yes . . . 5—Definitely not) adapted from Dufner et al. [49]; ($\alpha = 0.71$). 3. Nomophobia was measured on a 7-point Likert scale (1—Definitely do not agree . . . 7—Definitely agree) adapted from Yildirima and Correia [18]; ($\alpha = 0.74$). 4. Stress was measured on a 5-point Likert Scale (1—Never . . . 5—Always) adapted from Cohen [50]; ($\alpha = 0.78$) 5. Amount of sleep hours was measured on a 6-point Likert scale (1—Definitely do not agree . . . 6—Definitely agree) adapted from Tremblay et al. [51]; ($\alpha = 0.78$). 6. Problematic smartphone use was measured on a 6-point Likert scale (1—Definitely do not agree . . . 6—Definitely agree) adapted from Hong et al. [52]; ($\alpha = 0.75$).

*3.3. Procedure and Data Analysis*

SPSS™ software was used to perform data analysis at several stages. In the initial phase (Stage 1), both Study 1 and Study 2 data were analyzed with reliability tests for each variable and its corresponding items. A computation of new variables was performed as a mean value of the corresponding items, i.e., the Cronbach's alpha coefficient of internal consistency ("reliability") was calculated to make sure that the variables' Likert scales were reliable. Before conducting this analysis, it was important to ensure the consistency of all items with each other. To that end, a reverse score computation was performed for all the items that were "negatively phrased," thus transforming them into "new reversed" variables. Subsequently, the mean value of the corresponding items was computed. The computation was repeated twice for variables relating to COVID-19 (T1) and those after (T2). In addition, a Pearson two-tailed correlation to measure the relationship between the average explored variables was performed. In the second phase (Stage 2), an additional *t*-test was performed on the data of both Study 1 and Study 2, to evaluate whether there was a difference in PSU, loneliness, stress, and nomophobia values between subjects as measured for the two time periods. To understand the reasons for these changes (if found), in stage 3, a multiple hierarchical regression analysis was performed to evaluate the effect of the various independent variables on the dependent one (PSU). In stage 4, to verify the Study 2 research model, the analysis around the mediated model suggested by hypotheses H3 and H7, was framed. The analysis was performed using the Process macro for SPSS [53] to verify mediation effects based on a bootstrapping method of correlation bias; it was performed twice on the variables, for the two time periods T1 and T2.

**4. Results**

The descriptive analysis was first organized into categories to capture the level of each of the study's variables: loneliness, social affiliation, PSU, stress, sleep hours, and nomophobia. Tables 1 and 2 present descriptive statistics of the variables in the Study 1 and the Study 2 data.

**Table 1.** Means, standard deviation of the Study 1 variables (*N* = 197) for T1 and T2.

| Variable Name | T1; T2: Mean | T1; T2: SD |
|---|---|---|
| Loneliness | 4.24; 4.16 | 0.687; 0.675 |
| Stress | 2.83; 2.64 | 0.526; 0.728 |
| Sleep Hours | 2.51; 2.96 | 0.825; 0.763 |
| Nomophobia | 2.91; 2.96 | 0.727; 0.763 |
| Social Affiliation | 2.27; 2.21 | 0.727; 0.799 |
| PSU | 3.03; 2.85 | 0.725; 0.762 |

**Table 2.** Means, standard deviation of the Study 2 variables (*N* = 246) for T1 and T2.

| Variable Name | T1; T2: Mean | T1; T2: SD |
|---|---|---|
| Loneliness | 4.03; 3.91 | 0.718; 0.815 |
| Stress | 2.72; 2.91 | 0.611; 0.672 |
| Sleep Hours | 2.76; 2.57 | 0.863; 0.612 |
| Nomophobia | 2.99; 3.08 | 0.821; 0.848 |
| Social Affiliation | 2.57; 2.56 | 0.826; 0.887 |
| PSU | 3.15; 3.16 | 0.769; 0.860 |

### 4.1. Correlation Analysis

Correlations were computed on the Study 2 data (*n* = 246). The correlation analysis for the T1 and T2 periods of the study variables demonstrated a significant increase in the level of correlation between Social Affiliation and PSU during the period of T1 and T2 (*r* = 0.491 (T1), *r* = 0.539 (T2)). The correlation between Nomophobia and Social Affiliation decreased slightly from T1 (*r* = 0.493) to T2 (*r* = 0.488). The correlation between Nomophobia and PSU increased slightly in T2 (T1: *r* = 0.627, T2: *r* = 0.696). The correlation between Nomophobia and Sleep Hours increased slightly in T2 (T1: *r* = 0.317; T2: *r* = 0.363). Finally, the correlation between Sleep Hours and Stress decreased slightly in T2 (*r* = 0.286), as compared to T1 (0.385) (see Tables 3 and 4).

**Table 3.** Correlation analysis of the dependent variable (PSU) with the independent variables (T1) (Study 2 data).

| Variables | Mean | S.D. | 1 | 2 | 3 | 4 | 5 | 6 |
|---|---|---|---|---|---|---|---|---|
| 1. Loneliness | 3.73 | 0.51 | 1 | | | | | |
| 2. Social Affiliation | 2.50 | 0.91 | 0.09 | 1 | | | | |
| 3. PSU | 3.38 | 0.88 | 0.10 | 0.491 ** | 1 | | | |
| 4. Stress | 2.80 | 0.65 | −0.131 * | 0.218 ** | 0.235 ** | 1 | | |
| 5. Sleep Hours | 2.70 | 0.89 | −0.011 | 0.356 ** | 0.355 ** | 0.385 ** | 1 | |
| 6. Nomophobia | 3.04 | 0.94 | −0.106 | 0.493 ** | 0.627 ** | 0.273 ** | 0.317 ** | 1 |

Note: * $p < 0.05$, ** $p < 0.01$.

### 4.2. T-Test Analysis

A paired-samples *t*-test was conducted on both Study 1 and Study 2 data to compare the level of PSU in T1 and T2. The results indicated that there was a significant difference in the scores for Loneliness levels in T1 (*M* = 4.33, *SD* = 0.72) and T2 (*M* = 4.25, *SD* = 0.71); t(197) = 2.71, *p* < 0.01 attributed to Study 1. Similar results were obtained in Study 2 in T1 (*M* = 3.73, *SD* = 0.51) and T2 (*M* = 3.96, *SD* = 0.83); t(246) = −5.20, *p* < 0.01. In addition, there was a significant difference in the scores for Stress levels in T1 (*M* = 2.56, *SD* = 0.69) and T2 (*M* = 2.72, *SD* = 0.78); t(197) = −3.53, *p* < 0.01 attributed to Study 1. Similar results were obtained in Study 2 in T1 (*M* = 2.80, *SD* = 0.65) and T2 (*M* = 2.97, *SD* = 0.73); t(246) = −4.37, *p* < 0.01. Furthermore, there was also a significant difference in the scores for

Nomophobia levels in T1 ($M = 3.04$, $SD = 0.94$) and T2 ($M = 3.18$, $SD = 0.1.00$); t(246) = $-4.27$, $p < 0.01$ attributed to Study 2, while no significant difference was found in in T1 ($M = 2.88$, $SD = 0.86$) and T2 ($M = 2.93$, $SD = 0.92$); t(197) = $-1.43$, $p < 0.01$ attributed to Study 1. Overall, it was found that in most cases there was a significant difference in the variables' scores in T1 and T2 in both Study 1 and Study 2.

**Table 4.** Correlation analysis of the dependent variable (PSU) with the independent variables (T2) (Study 2 data).

| Variables | Mean | S.D. | 1 | 2 | 3 | 4 | 5 | 6 |
|---|---|---|---|---|---|---|---|---|
| 1. Loneliness | 3.96 | 0.84 | 1 | | | | | |
| 2. Social Affiliation | 2.45 | 0.98 | −0.147 * | 1 | | | | |
| 3. PSU | 3.33 | 0.95 | −0.076 | 0.539 ** | 1 | | | |
| 4. Stress | 2.98 | 0.74 | −0.333 ** | 0.121 | 0.181 ** | 1 | | |
| 5. Sleep Hours | 2.75 | 0.93 | −0.200 ** | 0.421 ** | 0.410 ** | 0.286 ** | 1 | |
| 6. Nomophobia | 3.18 | 1.00 | −0.194 ** | 0.488 * | 0.696 ** | 0.274 ** | 0.363 ** | 1 |

Note: * $p < 0.05$, ** $p < 0.01$.

### 4.3. Multiple Hierarchical Regression

Multiple hierarchical regression analysis was used to test whether the independent variables measured for the two time periods, T1 and T2, (loneliness, social affiliation, stress, sleep hours, and nomophobia) significantly predicted PSU. The Study 2 results related to the T1 period of the regression analysis indicated that four predictors: social affiliation, loneliness, sleep hours, and nomophobia explained 47.3% of the variance ($R^2 = 0.473$, F(246) = 6.192, $p < 0.01$). It was found that social affiliation significantly predicted PSU ($\beta = 0.194$, $p < 0.01$) as did nomophobia ($\beta = 0.507$, $p < 0.01$). It was also found that sleep hours ($\beta = 0.126$, $p < 0.05$) and loneliness predicted PSU ($\beta = 0.153$, $p < 0.05$). The results related to the T2 period of the regression analysis indicated that the same variables as found in T1 explained 55.6% of the variance ($R^2 = 0.556$, F(246) = 4.31, $p < 0.01$). Moreover, it was found that social affiliation significantly predicted PSU ($\beta = 0.226$, $p < 0.01$) as did nomophobia ($\beta = 0.556$, $p < 0.01$). It was also found that sleep hours ($\beta = 0.131$, $p < 0.01$) and loneliness ($\beta = 0.092$, $p < 0.05$) predicted PSU (Table 5).

**Table 5.** Results of the multiple hierarchical regression for T1 and T2.

| Variable Name | T1 | T2 |
|---|---|---|
| Loneliness | 0.153 * | 0.092 * |
| Sleep Hours | 0.126 * | 0.131 ** |
| Nomophobia | 0.507 ** | 0.556 ** |
| Social Affiliation | 0.194 ** | 0.226 ** |
| $R^2$ | 0.473 | 0.556 |
| F | 6.192 | 4.31 |
| Std, E | 0.644 | 0.64 |

Note: The results demonstrate the standardized beta coefficient (B). The dependent variable was PSU. * $p < 0.05$, ** $p < 0.01$.

### 4.4. Mediation Effect Analysis

Hypothesis 3 and Hypothesis 7 were tested using the Process macro for SPSS (Version 4), Model 4 [54,55]. The indirect effect was tested using a bootstrap estimation approach with 5000 samples [56].

Results related to Hypothesis 3: Social Affiliation will serve as a mediator between loneliness and PSU; it indicated that Loneliness was not found to be a significant predictor of Social Affiliation in T1 but was a significant predictor of Social Affiliation in

T2 (T1: B = 0.017, SE = 0.11, 95% CI [−0.21, 0.24], $\beta$ = 0.001, t = 0.147, $p > 0.05$; T2: B = −0.17, SE = 0.07, 95% CI [−0.32, −0.03], $\beta$ = −0.15, t = −2.33, $p < 0.05$).

Social Affiliation was found to significantly predict PSU in T1 and T2. T1: B = 0.47, SE = 0.05, 95% CI [0.37, 0.58], $\beta$ = 0.49, t = 8.81, $p < 0.01$; T2: B = 0.52, SE = 0.05, 95% CI [0.42, 0.63], $\beta$ = 0.53, t = 9.88, $p < 0.01$).

Loneliness was not found to significantly predict PSU in T1 and T2. T1: B = 0.16, SE = 0.96, 95% CI [−0.03, 0.35], $\beta$ = 0.09, t = 0.1.71, $p > 0.01$; T2: B = 0.00, SE = 0.62, 95% CI [−0.12, 0.13], $\beta$ = 0.01, t = 0.07, $p > 0.01$).

Approximately 25% of the variance was satisfactorily accounted for by the predictors ($R^2$ = 0.25) in T1 and 29% ($R^2$ = 0.29) in T2. The bootstrap results of the mediation test indicated that the mediation was positive but not significant in T1 and negative and significant in T2: (T1: B = 0.047, SE = 0.05, 95% CI [−0.037, 0.58]; T2: B = −0.09, SE = 0.04, 95% CI [−0.17, −0.02]) (see Table 6 and Figure 2).

**Table 6.** Mediation effect of Loneliness on Social Affiliation and PSU, Stress on Sleep Hours and PSU, and Nomophobia on Stress and PSU, and Social Affiliation on PSU for T1 and T2.

| Variable/Effect | T1: Coeff | T1: t | T1: $p$ | T2: Coeff | T2: t | T2: $p$ |
|---|---|---|---|---|---|---|
| Stress → PSU | 0.11 | 1.80 | 0.075 | 0.07 | 1.13 | 0.257 |
| Stress → Sleep Hours | 0.36 ** | 6.52 | 0.000 | 0.29 ** | 4.66 | 0.000 |
| Sleep Hours → PSU | 0.31 ** | 4.80 | 0.000 | 0.39 ** | 6.40 | 0.000 |
| Loneliness → PSU | 0.09 | 1.71 | 0.119 | 0.01 | 0.07 | 0.941 |
| Loneliness → Social Affiliation | 0.001 | 0.147 | 0.147 | −0.15 * | −2.33 | 0.021 |
| Social Affiliation → PSU | 0.49 ** | 8.81 | 0.000 | 0.53 ** | 9.88 | 0.000 |
| Nomophobia → Stress | 0.27 ** | 4.43 | 0.000 | 0.27 ** | 4.44 | 0.000 |
| Nomophobia → PSU | 0.60 ** | 12.56 | 0.000 | 0.70 ** | 14.61 | 0.000 |

Note: All path coefficients are standardized (* $p < 0.05$, ** $p < 0.01$).

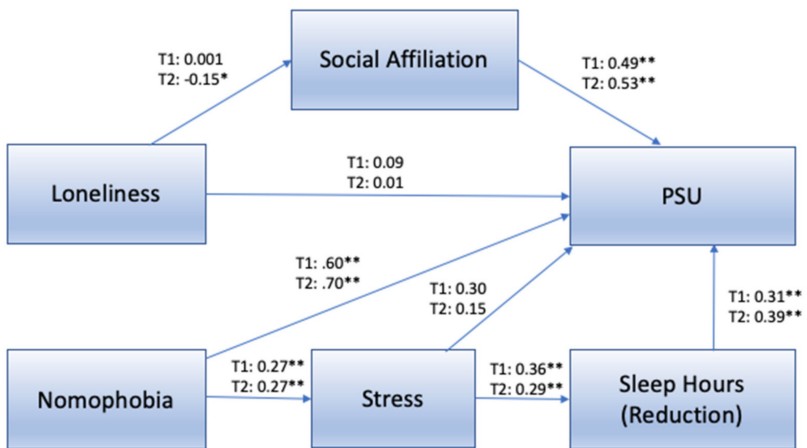

**Figure 2.** Relationships and mediation effects found for entire model [study data] in T1 and T2 (* $p < 0.05$, ** $p < 0.01$).

Study 2 results related to Hypothesis 7: Sleep hours will serve as a mediator between stress and PSU; it indicated that stress significantly predicted reduction in sleep hours in both T1 and T2: (T1: B = 0.53, SE = 0.08, 95% CI [0.37, 0.69], $\beta$ = 0.36, t = 6.52, $p < 0.01$; T2: B = 0.36, SE = 0.07, 95% CI [0.20, 0.51], $\beta$ = 0.29, t = 4.66, $p < 0.01$).

The Sleep Hours variable was found to significantly predict PSU in T1 and in T2, (T1: B = 0.31, SE = 0.06, 95% CI [0.18, 0.43], $\beta$ = 0.31, t = 4.80, $p < 0.01$. T2: B = 0.40, SE = 0.06, 95% CI [0.27, 0.52], $\beta$ = 0.39, t = 6.40, $p < 0.01$).

Moreover, Stress was found to have a positive relationship with PSU in T1 and T2 (T1: B = 0.36, SE = 0.09, 95% CI [0.18, 0.53], $\beta$ = 0.30, t = 4.12, $p < 0.01$; T2: B = 0.19, SE = 0.09, 95% CI [0.002, 0.380], $\beta$ = 0.15, t = 1.99, $p < 0.01$). However, when the equation was controlled by sleep hours, Stress was not found to be significantly related to PSU in T1

and T2, (T1: B = 0.21, SE = 0.09, 95% CI [0.03, 0.39], $\beta$ = 0.17, t = 2.38, $p$ > 0.01; T2: B = 0.11, SE = 0.09, 95% CI [$-$0.07, 0.30], $\beta$ = 0.09, t = 1.19, $p$ > 0.01). Therefore, these results indicate that the mediation effect was found to be significant in both T1 and T2.

Approximately 15% of the variance was accounted for by the predictors ($R^2$ = 0.15) in T1 and 17% ($R^2$ = 0.17) in T2, respectively. The bootstrap results of the mediation test indicated that the mediation was positive and significant in T1 and in T2. The indirect coefficient was positive and significant in T1 and in T2 (T1: B = 0.16, SE = 0.04, 95% CI [0.08, 0.25]); T2: B = 0.14, SE = 0.04, 95% CI [0.07, 0.23]).

Hence, Hypothesis 7 was supported in T1 and T2 (see Table 6 and Figure 2).

Finally, the mediation effect analysis on the Study 2 data was confirmed by the regression analysis for the following variables/effects: Nomophobia $\rightarrow$ PSU; Social Affiliation $\rightarrow$ PSU; Sleep Hours $\rightarrow$ PSU.

A summary of Study 2's main findings is shown in Table 7.

**Table 7.** Summary of hypotheses and main regression analysis findings (Study 2). (The regression analysis results demonstrate the standardized beta coefficient. The dependent variable was PSU. * $p$ < 0.05, ** $p$ < 0.01.)

| Hypothesis Number | Hypothesis | T1 | T2 |
|---|---|---|---|
| H1 | Loneliness is positively associated with social affiliation | Not Supported | Not Supported |
| H2 | Desire for greater social affiliation is positively associated with PSU | Supported | Supported |
| H3 | Social affiliation will serve as a mediator between loneliness and PSU | Not Supported | Supported |
| H4 | Nomophobia is positively associated with PSU | Supported | Supported |
| H5 | Mobile users with fewer sleep hours will exhibit a higher level of PSU | Supported | Supported |
| H6 | Stress level will positively correlate with number of sleep hours | Supported | Supported |
| H7 | Sleep hours will serve as a mediator between stress and PSU | Supported | Supported |
| **Multiple Hierarchical Regression Analysis** | | | |
| **Variable Name** | **T1** | | **T2** |
| Loneliness | 0.153 * | | 0.092 * |
| Sleep Hours | 0.126 * | | 0.131 ** |
| Nomophobia | 0.507 ** | | 0.556 ** |
| Social Affiliation | 0.194 ** | | 0.226 ** |
| **Mediation Effect Analysis** | | | |
| **Variable/Effect** | **T1: Coeff** | | **T2: Coeff** |
| Stress $\rightarrow$ Sleep Hours | $\beta$ = 0.36 (T1) $p$ < 0.01 | | $\beta$ = 0.29 (T2) $p$ < 0.01 |
| Sleep Hours $\rightarrow$ PSU | $\beta$ = 0.31 (T1) $p$ < 0.01 | | $\beta$ = 0.39 (T2) $p$ < 0.01 |
| Social Affiliation $\rightarrow$ PSU | $\beta$ = 0.49 (T1) $p$ < 0.01 | | $\beta$ = 0.53 (T2) $p$ < 0.01 |
| Nomophobia $\rightarrow$ PSU | $\beta$ = 0.60 (T1) $p$ < 0.01 | | $\beta$ = 0.70 (T2) $p$ < 0.01 |
| Loneliness $\rightarrow$ Social Affiliation | $\beta$ = 0.001 (T1) $p$ > 0.05 | | $\beta$ = $-$0.15 (T2) $p$ < 0.05 |

## 5. Discussion

This study aimed to evaluate how problematic smartphone use was influenced by several factors: nomophobia, stress, reduction in sleeping hours, social affiliation, and sleeping hours during two time periods: 1. T1—the first COVID-19 lockdowns in Israel

and 2. T2—after the first COVID-19 lockdowns ended. The study results indicated that social affiliation, nomophobia, and sleep hours are the main predictors that explain PSU in both T1 and T2 among the study participants. Moreover, sleep hours were found to mediate between stress and PSU in T2 and in T1 (H7). However, although the effect of the level of stress on the reduction in sleep hours was lower in T2 than in T1, thus allowing more hours of sleep than in T1, it was found that the effect on the PSU level was higher (T1: $\beta = 0.31$ vs. T2: $\beta = 0.39$). This may be explained by the decrease in the level of stress in T2, after the COVID-19 lockdown ended, but the enduring implications of problematic smartphone usage might have created an increase in the addiction level of smartphone use among the respondents even though their quality of sleep was better once COVID-19 was less widespread. This finding may also be explained and supported by the concept of moral feelings [57]. According to this concept, moral distress is defined as a situation in which individuals know how they should act but are not able to pursue the right course of action. Moreover, Jameton [58] defined moral distress as a psychological disequilibriuum and a negative feeling state that prevent individuals from performing the right behavior that follows a moral decision. Such a situation may become visible in conflict situations. Therefore, although according to the current study respondents showing a reduced level of stress and better-quality sleep hours in T2 were expected to exhibit a reduced PSU behavior, the slightly higher levels of PSU actually found in T2 may express their inability to perform the anticipated behavior. This finding was also reinforced by the difference in the effect of social affiliation levels in T1 compared to T2. As the COVID-19 lockdown ended, respondents were free of obstacles to meeting other people; thus, in T2, with the increase in sleep hours together with the reduction in stress positively influencing the amount of sleep hours (improving sleep quality), a reduction in the level of PSU among the respondents might have been expected. However, the effect of sleep hours along with the effect of social affiliation on PSU was found higher in T2 as compared to T1. This finding might be explained by the continued negative impact of the COVID-19 lockdown on the level of addiction among the respondents, hence their excessive smartphone usage in T2 as compared to T1. This explanation may also be supported and justified by Zhao and Zhou [15], who emphasized the negative long-run implications of COVID-19 lockdowns on the level of addiction among the respondents. Moreover, in T2, the effect of Social Affiliation on PSU (H2) was stronger, as compared to Sleep Hours on PSU. This finding was obtained through the evaluation of the regression results (T2 Social Affiliation: $\beta = 0.27$ vs. T2 Sleep Hours: $\beta = 0.13$). Finally, the findings are also supported and intensified by the Nomophobia effect on PSU (H7). As the fear of losing smartphone access in T2 increased, as compared to T1 (T1: $\beta = 0.60$ vs. T2: $\beta = 0.70$), the level of PSU in T2 was much higher than in T1. These findings may also point to the negative and enduring addiction outcomes that influenced young adults when COVID-19 was less widespread.

The study's findings are consistent with current and earlier studies that found that PSU and nomophobia are strongly connected as hypothesized in H4 [44,59]. In addition, as in previous studies, it was found that PSU is associated with the need for social media communication [60]. The current study is consistent with Carbonell et al.'s study as it shows that problematic smartphone use is influenced by circumstance; during T2 it was more influenced by social affiliation (T1: $\beta = 0.49$ vs. T2: $\beta = 0.53$) and by the increasing level of Sleep Hours—i.e., less reduction in sleeping hours (T1: $\beta = 0.31$ vs. T2: $\beta = 0.39$). PSU was also influenced by Nomophobia (T1: $\beta = 0.60$ vs. T2: $\beta = 0.70$). In other words, the study findings show that the impact of nomophobia on PSU was not decreased in T2 than in T1 as expected but rather the opposite. With adjustment to the decrease in the level of stress in T2 compared with the level of stress in T1 along with the increased impact of social affiliation on PSU in T2, when respondents were able to meet other people face to face without obstacles, it was expected to receive a decrease in the levels of PSU in T2 than in T1; however, on the contrary, the level of PSU obtained in T2 was higher than in T1. This result can be explained by the fact that in T1, subjects were starting to develop connections through social media, already during or before T1; therefore, when the level of

stress decreased slightly in T2 ($\beta = 0.36$ vs. $\beta = 0.29$), it may be that the fear of not being able to develop and maintain those connections via social media was increased, thus leading to high levels of smartphone addiction. The reason why loneliness was not associated with social affiliation (H1) in T1 and PSU in both T1 and T2 may be explained by the fact that the average age of the participants was relatively young (mean age = 27), and, therefore, most of the subjects were already active on social media for years prior to the first COVID-19 outbreak; thus, its overall influence, especially in T1 (as COVID-19 emerged) is close to negligible. However, as mentioned by Enez et al. [23], loneliness serves as a predictor of psychological symptoms such as depression and stress; therefore, it might be reasonable to infer that the level of stress that actually affected the level of PSU, especially in T1, embodied young people's feeling of loneliness during the COVID lockdowns. Moreover, several studies (e.g, Skues et al. [25]) show that loneliness is not always found to be a significant predictor of PSU. Some studies (for instance, Lee et al. [61]) attribute PSU to the feeling of being affiliated with other people that provides a sense of belonging, which may lower the impact of loneliness or completely nullify its importance for PSU. An additional explanation for loneliness not acting as a predictor of PSU may be inferred from an inadequate interpretation of this emotional state. For example, while early studies considered loneliness to be composed of only one dimension [62], later studies showed that loneliness is associated with two dimensions: 1. Social loneliness, which derives from the lack of close relationships with others, and 2. Emotional loneliness, which derives from not interacting in social circles [63,64]. Both social and emotional loneliness may result in negative feelings about oneself, such as depression and sadness [65], which may trigger individuals to use their mobile phones in an excessive manner as a means of building relationships, satisfying their need to belong [66,67], and reducing the negative feelings that arise from difficult situations [65]. Finally, the current study's finding regarding loneliness not being directly related to PSU is also supported by Takao et al. [68] and Skues [25]. This relationship, or lack thereof, requires further exploration in future studies. The entire model and its correlations are shown in Figure 2.

## 6. Conclusions

The study enhances the understanding and implications of the COVID-19 outbreak on PSU by investigating various important variables. The study's implications may especially be important for practitioners who strive to improve their knowledge and understanding related to risk factors that influence or spur PSU. In addition, the study may assist in the development of successful intervention programs that aim to reduce the level of PSU among addicted young adults. Moreover, the study emphasizes the need for adequate control of social media usage and its undesirable side effects such as stress and behavioral disorders.

## 7. Limitations and Further Research

The study did not evaluate additional variables that may be associated with PSU. It is suggested that future studies extend the model as well as the sample size and group of subjects and test additional variables, including psychological variables [69]. However, the sample size and further analysis conducted on the Study 1 data (see Appendix A) may assist scholars in extending this study through the utilization of the findings and the substantial number of respondents in both studies ($n = 443$). It is also suggested to explore further the bidirectional relationship between stress and addiction as part of the proposed model. Following Buchanan et al. [70], PSU may be seen as a reason for people to escape from difficult situations and reduce stress; however, a deficient stress control mechanism may lead to an increased risk of addictive behavior involving smartphones.

**Funding:** This research received no external funding. The APC was funded by the Research and Development Authority at Ariel University.

**Institutional Review Board Statement:** The study was approved by the Ethics Committee of Ariel University (Protocol Code: AU-SOC-MZ-20211108; Date of Approval: 8 November 2021) for studies involving humans.

**Informed Consent Statement:** A written informed consent was obtained from all subjects involved in the study.

**Data Availability Statement:** The data presented in this study are available on request from the corresponding author.

**Conflicts of Interest:** The author declares no conflict of interest.

## Appendix A

**Table A1.** Correlation analysis of the dependent variable (PSU) with the independent variables (T1) (Study 1 data).

| Variables | Cronbach Alpha | Mean | s.d. | 1 | 2 | 3 | 4 | 5 | 6 |
|---|---|---|---|---|---|---|---|---|---|
| 1. Loneliness | 0.84 | 4.24 | 0.68 | 1 | | | | | |
| 2. Social Affiliation | 0.71 | 2.27 | 0.73 | 0.075 | 1 | | | | |
| 3. PSU | 0.75 | 3.03 | 0.72 | 0.103 | 0.488 ** | 1 | | | |
| 4. Stress | 0.78 | 2.83 | 0.53 | −2.39 ** | 0.213 ** | 0.257 ** | 1 | | |
| 5. Sleep Hours | 0.68 | 2.51 | 0.82 | −0.163 * | 0.427 ** | 0.210 ** | 0.375 ** | 1 | |
| 6. Nomophobia | 0.72 | 2.91 | 0.73 | 0.048 | 0.462 ** | 0.605 ** | 0.223 ** | 0.277 ** | 1 |

Note: * $p < 0.05$, ** $p < 0.01$.

**Table A2.** Correlation analysis of the dependent variable (PSU) with the independent variables (T2) (Study 1 data).

| Variables | Cronbach Alpha | Mean | s.d. | 1 | 2 | 3 | 4 | 5 | 6 |
|---|---|---|---|---|---|---|---|---|---|
| 1. Loneliness | 0.83 | 4.16 | 0.67 | 1 | | | | | |
| 2. Social Affiliation | 0.81 | 2.21 | 0.79 | −0.048 | 1 | | | | |
| 3. PSU | 0.74 | 2.85 | 0.76 | 0.021 | 0.595 ** | 1 | | | |
| 4. Stress | 0.77 | 2.64 | 0.73 | −0.233 ** | 0.187 ** | 0.198 ** | 1 | | |
| 5. Sleep Hours | 0.74 | 2.74 | 0.56 | −0.292 ** | 0.478 ** | 0.369 ** | 0.374 ** | 1 | |
| 6. Nomophobia | 0.78 | 2.96 | 0.76 | 0.041 | 0.442 ** | 0.652 ** | 0.222 ** | 0.296 ** | 1 |

Note: ** $p < 0.01$.

**Table A3.** Results of the multiple hierarchical regression for T1 and T2 (Study 1 data).

| Variable Name | T1 | T2 |
|---|---|---|
| Loneliness | −0.060 | 0.019 |
| Stress | 0.099 ** | 0.021 |
| Sleep Hours | −0.738 | 0.057 |
| Nomophobia | 0.482 ** | 0.484 ** |
| Social Affiliation | 0.265 ** | 0.381 ** |
| $R^2$ | 0.421 | 0.537 |
| F | 18.431 | 49.457 |
| Std, E | 0.665 | 0.625 |

Note: The results demonstrate the standardized beta coefficient (B). The dependent variable was PSU. ** $p < 0.01$.

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
