# Peer review of "The Impact of Nomophobia, Stress, and Loneliness on Smartphone Addiction among Young Adults during and after the COVID-19 Pandemic: An Israeli Case Analysis"

_sustainability, doi:10.3390/su14063229_

Round 1
Reviewer 1 Report
Excellent contribution to understand the transformation of social patters and the impact of nomophobia, stress and loneliness on smartphone addiction among young adults in two key periods during and after COVID-19 lockdown. The psychological influences seem very important to me; loneliness, stress and nomophobia and the behavioral influences; desaire to belong to a social community, lack of a sufficient number os sleep hours and problematic smartphone use (PSU) and smartphone addiction, as well as the modern disorders as a "phubbing". the methodology seems very complete; its base of theorical references offers us a very detailed overview of the research that han been published in this regard. the conclusions presented seem to me to be broad and detailed and will undoubtedly serve as a reference to future research. I agree with the author, important for practitiones who strive to improve their knowledge and understanding related to risk factors that influence or stimulate PSU. In addition, the study may assist in the development of successul interventios programs that aim to reduce the level of PSU. Moreover, the study emphasizes the need for adequate control of social media usage and its undesirable side effects such a stress and behavioural disorders. Congratulations !
Author Response
Reply to Reviewer 1's Comments
Revision of: The impact of nomophobia, stress, and loneliness on smartphone addiction among young adults during and after the Covid-19 pandemic: An Israeli case analysis (Ref: Sustainability, Manuscript ID -1596044)
I am grateful to Reviewer 1 for his/her careful reading of my paper and his/her detailed comments. Below are the Reviewer’s comments followed by my responses.
Excellent contribution to understand the transformation of social patters and the impact of nomophobia, stress and loneliness on smartphone addiction among young adults in two key periods during and after COVID-19 lockdown. The psychological influences seem very important to me; loneliness, stress and nomophobia and the behavioral influences; desire to belong to a social community, lack of a sufficient number of sleep hours and problematic smartphone use (PSU) and smartphone addiction, as well as the modern disorders as a "phubbing". the methodology seems very complete; its base of theorical references offers us a very detailed overview of the research that has been published in this regard. the conclusions presented seem to me to be broad and detailed and will undoubtedly serve as a reference to future research. I agree with the author, important for practitioners who strive to improve their knowledge and understanding related to risk factors that influence or stimulate PSU. In addition, the study may assist in the development of successful interventions programs that aim to reduce the level of PSU. Moreover, the study emphasizes the need for adequate control of social media usage and its undesirable side effects such a stress and behavioral disorders. Congratulations!
Reply: I thank the Reviewer and appreciate his/her positive comments.

Reviewer 2 Report
The paper “The impact of nomophobia, stress, and loneliness on smartphone addiction among young adults during and after the Covid-19 pandemic: An Israeli case analysis” reviews the important and timely topic of addictive behaviors and outcomes during and after the Israeli government lockdown in early 2020. The authors pose a number of interesting questions, and use multiple statistical strategies to answer them. Here are some suggestions for ways to strengthen this paper:
- The figures and tables are missing, so it is hard to evaluate some aspects of this paper.
- Additional editing of the paper could reduce overlap and redundancy (e.g., defining the acronym for PSU several times, defining the term nomophobia several times, wordiness in general).
- The term “theoretical basis” and theory should be replaced with “conceptual framework” as this study appears to lack a theoretical basis.
- Nomophobia does not appear explicitly appear in the DSM-IV, as is stated on p. 4
- It is not appropriate to test sleep as a mediator between stress and PSU, as there is no direct effect observed in the multiple regression between stress and PSU to be explained. I recommend reviewing this source for more information: https://davidakenny.net/cm/mediate.htm#:~:text=The%20way%20to%20measure%20mediation,1%20%2D%20c'%2Fc.
- It may be helpful to use the term “construct” or “scale” when referring to loneliness, social affiliation, nomophobia, stress, sleep hours, and problematic smartphone use.
- I would like to know how the questions/measures for loneliness, social affiliation, nomophobia, stress, sleep hours, and problematic smartphone use were adapted from the original (perhaps this is clear in the tables or appendices, but since they are missing I have no way to tell).
- It would be good to add the range of possible scores for the loneliness, social affiliation, nomophobia, stress, sleep hours, and problematic smartphone use scales.
- Why do you distinguish between pretest and study data? Why not call it “sample 1” and “sample 2” (or “study 1” and “study 2”)?
- The text should not repeat the table content. It would be better to highlight some significant relationships, but then present the statistical details in the tables (but not the text).
- The meaning of what was done during Stage 1 (see lines 227-231) of the data analyses is unclear; please clarify your methodology.
- Computing the descriptive statistics seems to be “stage 1” for the analyses
- Presumably you conducted a series of paired samples t-tests (not just one t-test)
- Lines 329-338 seem out of place, since these results are not related to hypothesis 3 or 7.
Author Response
Reply to Reviewer 2's Comments
Revision of: The impact of nomophobia, stress, and loneliness on smartphone addiction among young adults during and after the Covid-19 pandemic: An Israeli case analysis (Ref: Sustainability, Manuscript ID -1596044)
I am grateful to Reviewer 2 for his/her careful reading of my paper and his/her detailed comments. Below are the Reviewer’s comments followed by my responses.
General Comments: The paper “The impact of nomophobia, stress, and loneliness on smartphone addiction among young adults during and after the Covid-19 pandemic: An Israeli case analysis” reviews the important and timely topic of addictive behaviors and outcomes during and after the Israeli government lockdown in early 2020. The authors pose a number of interesting questions and use multiple statistical strategies to answer them. Here are some suggestions for ways to strengthen this paper:a number of interesting questions and use multiple statistical strategies to answer them. Here are some suggestions for ways to strengthen this paper:
- The figures and tables are missing, so it is hard to evaluate some aspects of this paper.
Reply: Thank you for this constructive comment. I really apologize for the inconvenience. Unfortunately, it was not possible (I assume due to technical issues) to upload the figures and tables to the journals’ website during the submission process. This forced me to send them separately to the journal’s branch office. I understand from your comment that you did not receive the files. Therefore, I enclose them with the submission of this revised manuscript. I do hope that this time you will receive them properly.
- Additional editing of the paper could reduce overlap and redundancy (e.g., defining the acronym for PSU several times, defining the term nomophobia several times, wordiness in general).
Reply: Thank you for this constructive comment. Overlaps and redundant acronyms have been reduced throughout the manuscript, especially after the first introduction of the definition of the terms.
- The term “theoretical basis” and theory should be replaced with “conceptual framework” as this study appears to lack a theoretical basis.
Reply: Thank you for this constructive comment. The terms was replaced as suggested. Please refer to lines 82 and 221.
- Nomophobia does not appear explicitly appear in the DSM-IV, as is stated on p. 4
Reply: Thank you for this constructive comment. The Reviewer is right, the term does not appear either in DSM-IV or in DSM-V, although it was already proposed to be included in the list of anxiety categories [https://www.ncbi.nlm.nih.gov/pmc/articles/PMC4036142/]. Following the Reviewer’s comment, the phrase “This definition was included as a psychological disorder in the American DSM-IV handbook” was changed to “This definition has been proposed for inclusion as a psychological disorder in the fifth edition of the American Diagnostic and Statistical Manual of Mental Disorders (DSM-5)”. Please refer to lines 125-127.
- It is not appropriate to test sleep as a mediator between stress and PSU, as there is no direct effect observed in the multiple regression between stress and PSU to be explained. I recommend reviewing this source for more information: https://davidakenny.net/cm/mediate.htm#:~:text=The%20way%20to%20measure%20mediation,1%20%2D%20c'%2Fc.
Reply: Thank you for this constructive comment and for the useful reference.
I agree with the Reviewer’s comment that the mediation effect was not completely displayed. After reviewing the reference provided by the Reviewer and observing the multiple regression between stress and PSU obtained by multiple stepwise regression analysis and the SPSS process v.4.0 output, which was re-executed, I can offer the following explanation: 1. A positive significant direct relationship between stress and PSU was found without the control of sleep. 2. Stress was found to be significantly related to the reduction in sleep hours 3. Sleep Hours were found to be significantly related to PSU. 4. Regarding the regression analysis, it was executed in a stepwise method; therefore, the stress variable was excluded because of multicollinearity with sleep hours and/or PSU. This result, along with the mediation results obtained by the PROCESS macro for SPSS v.4.0, strengthens the full mediation of sleep hours on the relationship between stress with PSU. Please refer to lines 802-807 for details on how I completed the missing information regarding the mediation that was ultimately found to be significant.
- It may be helpful to use the term “construct” or “scale” when referring to loneliness, social affiliation, nomophobia, stress, sleep hours, and problematic smartphone use.
Reply: Thank you for this constructive comment. The term “construct” was added in line 244 where the variables, instrument, and measured are described. The term “scales” is already mentioned in the text in line 246, where for each variable the “adapted” scale from the literature is cited. (please refer to the following: “… for each evaluated variable the following scales were adapted” - line 246).
- I would like to know how the questions/measures for loneliness, social affiliation, nomophobia, stress, sleep hours, and problematic smartphone use were adapted from the original (perhaps this is clear in the tables or appendices, but since they are missing, I have no way to tell).
Reply: Thank you for this comment. The questions were taken “as is” from the literature and were translated into the Hebrew language. To make sure that they were still valid, they were rechecked back by professionals to confirm their full correspondence with the original items and scales. A clarification regarding this issue was added in lines 242-244.
- It would be good to add the range of possible scores for the loneliness, social affiliation, nomophobia, stress, sleep hours, and problematic smartphone use scales.
Reply: Thank you for this constructive comment. The scales for all variables were already presented (according to the literature) in the original text. Please refer to lines 243–244;247-255.
Loneliness was measured on a 4-point Likert scale (1 – Never … 4 – Always) adapted from Russell [42]; (α = .84). 2. Social affiliation was measured on a 5-point Likert scale (1 – Definitely yes … 5 – Definitely not) adapted from Dufner et al., [43]; (α = .71). 3. Nomophobia was measured on a 7-point Likert scale (1 - Definitely do not agree … 7- Definitely agree) adapted from Yildirima and Correia [18]; (α = .74). 4. Stress was measured on a 5-point Likert Scale (1 – Never … 5 – Always) adapted from Cohen [44]; (α = .78) 5. Amount of sleep hours was measured on a 6-point Likert scale (1 – Definitely do not agree … 6 – Definitely agree) adapted from Tremblay et al., [45]; (α = .78). 6. Problematic smartphone use was measured on a 6-point Likert scale (1 – Definitely do not agree … 6 – Definitely agree) adapted from Hong et al. [46]; (α = .75).”
Nevertheless, in order to err on the side of caution and to provide the required clarifications regarding the use ofnumeric scales coded in accordance with the existing literature, the following text was added in line 243: “and the original scales were coded respectively according to their format in the literature”
- Why do you distinguish between pretest and study data? Why not call it “sample 1” and “sample 2” (or “study 1” and “study 2”)?
Reply: I greatly appreciate this comment. The Reviewer is right, there is no need to distinguish between pretest and study data. Following this comment, the pretest (exploratory) stage has been renamed “Study 1” and the confirmatory study is now called “Study 2”. It should be clarified that since the first questionnaire (Study 1) was initially distributed to a convenience sample (students), it was considered to provide exploratory data that required confirmation. In order to obtain such confirmation, the questionnaire was administered again to a sample recruited from the general public, and this process is known as Study 2. (Please refer to lines 206-209).
- The text should not repeat the table content. It would be better to highlight some significant relationships, but then present the statistical details in the tables (but not the text).
Reply: I greatly appreciate this comment. I have deleted some of the redundant and repeated text in places where the statistical details are easy to interpret and can be understood directly from the tables (such as in section 4.1 – “Correlation analysis”; refer to lines 520-528). However, in places such as section 4.4 (“Mediation effect analysis”), where only the main outcomes of the model are reported in the tables (for example, Table 6), the text was left as it was with the addition of supporting details.
- The meaning of what was done during Stage 1 (see lines 227-231) of the data analyses
is unclear; please clarify your methodology.
Reply: I greatly appreciate this comment. A clarification of this stage was provided in the revised text. It was explained that before conducting the reliability test to measure internal consistency (Cronbach’s alpha), all items that were “negatively phrased” were recoded into new variables (referred to as “new-variable-Reversed”). (Refer to lines 260-266).
- Computing the descriptive statistics seems to be “stage 1” for the analyses
Presumably you conducted a series of paired samples t-tests (not just one t-test)
Reply: I greatly appreciate this comment. Following the Reviewer’s comment, in addition to the t-test analysis performed on data attributed to Study 2, a further t-test analysis was performed on the data attributed to Study 1. (Refer to lines 268-269; 533-755).
- Lines 329-338 seem out of place, since these results are not related to hypothesis 3 or 7.
Reply: I highly appreciate this comment. These results were obtained while testing the whole model, but since they are not related to either hypothesis 3 or 7, they have been deleted from the text as suggested by the Reviewer (refer to line 815).

Reviewer 3 Report
The article is original because it shows the personality functioning of the respondents during a pandemic, measured in a time interval. This mainly concerns the dimension of the impact of stress. The introduction is adequately drawn up on the basis of contemporary literature on nomophobia and ends with the formulation of hypotheses. I only have reservations about the discussion part of the article that it does not refer enough to hypotheses and explain little based on the concepts of stress and loneliness.
Author Response
Reply to Reviewer 3's Comments
Revision of: The impact of nomophobia, stress, and loneliness on smartphone addiction among young adults during and after the Covid-19 pandemic: An Israeli case analysis (Ref: Sustainability, Manuscript ID -1596044)
I am grateful to Reviewer 3 for his/her careful reading of my paper and his/her detailed comments. Below are the Reviewer’s comments followed by my responses.
The article is original because it shows the personality functioning of the respondents during a pandemic, measured in a time interval. This mainly concerns the dimension of the impact of stress. The introduction is adequately drawn up on the basis of contemporary literature on nomophobia and ends with the formulation of hypotheses. I only have reservations about the discussion part of the article that it does not refer enough to hypotheses and explain little based on the concepts of stress and loneliness.
Reply: I thank the Reviewer for this suggestion. Based on the Reviewer’s comments, the Discussion section was further elaborated to include additional references to the study’s hypotheses and better explain the concepts of stress and loneliness. (Refer to lines 831;973-975;979;984;1009-1013)

Round 2
Reviewer 2 Report
This very of the paper is much stronger. The author(s) have done a good job of responding to most of my concerns. My only remaining suggestion relates to the lack of a table; I still believe that creating a table to report some of the results (rather than embedding all of the statistical details in the text) would be more efficient, however.
Author Response
Reply: Thank you for this constructive comment. Based on the Reviewer’s comments, A table (Table 7) that summarizes the main results is enclosed. (Refer to lines 357–558).
Reviewer 3 Report
the article is slightly improved, but I would still expect a more in-depth discussion of the results, referring to the theory of stress and the theory of loneliness
The work is correctly structured in terms of methodology. The introduction adequately introduces the research problem, and the research tools and the method of conducting the research are correct. The discussion of the results, which largely repeats the presented research, raises doubts. And little refers to a broader theoretical background, e.g. the concept of moral feelings or the concept of feelings in difficult situations.
Author Response
Reply: Thank you for this constructive comment. Based on the Reviewer’s comments, several references (relating mainly to loneliness and stress) were added to the Literature Review section (refer to lines 115–124; 232–236). In addition, the Discussion section was improved and better connected to the results. References to studies on stress, loneliness, and addiction that were used to support and explain the findings were also included in this section (refer to lines 675-689). Moreover, the theoretical background was broadened to encompass the issue of moral feelings (refer to lines 578–587). Finally, several new additional suggestions for future work were added to improve and strengthen the paper’s quality (refer to lines 913-916).
